# Impacts of Physical Exercise and Media Use on the Physical and Mental Health of People with Obesity: Based on the CGSS 2017 Survey

**DOI:** 10.3390/healthcare10091740

**Published:** 2022-09-11

**Authors:** Han Wang, Yang Yang, Qingqing You, Yuwei Wang, Ruyue Wang

**Affiliations:** 1School of Journalism and Communication, Jinan University, Guangzhou 510632, China; 2School of Journalism and Communication, Beijing Sport University, Beijing 100084, China; 3School of Journalism and Communication, Nanjing Normal University, Nanjing 210098, China

**Keywords:** physical exercise, media use, obesity, physical health, mental health

## Abstract

Obesity has become a common chronic disease in many countries around the world. People with obesity, as a minority, deserve more social attention. Currently, there are few studies on the health level of people with obesity from the perspective of social influencing factors. This study examines the effects of the frequency of physical exercise and frequency of use of different media types on the level of physical and mental health in people with obesity. In addition, we explore the mediating effect of physical exercise frequency on the relationship between online media use frequency and the mental health of people with obesity. The participants are 882 patients with obesity from the 2017 China General Social Survey (CGSS). The results show that: (1) Online media use was a possible positive predictor of physical health improvement among people with obesity. (2) Physical exercise was a possible positive predictor of mental health improvement among people with obesity. (3) Physical exercise played an entirely mediating role between online media use and mental health. The study is conducive to understanding the relationship and mediating mechanism between physical exercise, media use, and physical and mental health in people with obesity. The results of this study can provide suggestions for improving the health and well-being of people with obesity. Future research could explore more aspects of physical activity (e.g., the methods for physical exercise) and media use (e.g., media preference or compulsive use). More variables can be included in the study of influencing factors of the physical and mental health of people with obesity.

## 1. Introduction and Literature Review

Obesity has become a severe public health problem in the world. The latest data released by the World Health Organization (WHO) show that more than 1 billion people are obese globally, which is still increasing [1]. Similarly, the growing obese population also means that China is facing an obesity crisis. The incidence and growth rate of obesity in China ranks first globally, and China has the largest number of people with obesity in the world [2]. In the context of the high obesity rate in China, the size of obesity groups has made them a group that cannot be ignored by society.

Obesity is a state of excessive body fat accumulation that may harm health. Body mass index (BMI, BMI = kg/m^2^) is usually used as the defining standard of obesity [1]. The WHO defines obesity as a disease [3]. According to research on the effects of obesity on health, those with obesity had significantly higher blood pressure, blood lipid levels, blood sugar, and other measurements than those with normal BMI ranges. Additionally, hypertension and hyperlipidemia have been shown to be important risk factors for cardiovascular and cerebrovascular diseases such as coronary heart disease and stroke [4]. Obesity adversely affects the health of people with obesity and affects their mental health to some extent. Obesity can easily cause a variety of psychological problems. Depression, anxiety, low sense of happiness and satisfaction, interpersonal sensitivity, and other psychological issues are more evident in groups with obesity [5]. Therefore, attention to the physical and mental health of the obese group is an important issue worthy of study.

In previous studies on the influencing factors of public health, media use and physical exercise are two important factors affecting people’s physical and mental health. On the one hand, media is an important channel for the public to receive information. With the continuous development of internet technology, online media plays an increasingly important role in people’s lives. Research shows that media use can ultimately affect health outcomes by influencing people’s health beliefs and behaviors [6]. On the other hand, empirical research shows that moderate physical exercise can help improve physical health, alleviate people’s anxiety, depression, and other problems, and improve mental health [7]. Obesity is a complex health condition caused by behavioral, genetic, environmental, and physiological factors [8]. Compared with the healthy group, the influencing factors of physical and mental health in the obese group may be more complex. Therefore, under the realistic background that the particular group of people with obesity accounts for a large proportion of society, it is necessary to investigate the possible influence mechanism of media use and physical exercise on the physical and mental health of people with obesity.

### 1.1. Impacts of Media Use on the Health of People with Obesity

Studies have suggested that media use may be one of the causes of rising obesity rates. Research shows that the longer the media is used, the greater the risk of obesity [9]. One possible explanation is that the increase in media use has resulted in mass sedentary behavior for a long time and a significant reduction in physical exercise time [10]. Many studies have examined the correlation between media use and obesity. Media use, whether traditional media [11] or online media [12], can lead to a decrease in physical activity and an increase in sedentary behavior, increasing the risk of obesity. The increasing risk of obesity indicates the further deterioration of the health and mental health of people with obesity [13]. In addition, since the media content will affect people’s cognition, thereby affecting people’s attitudes and behaviors [14], the content of media reports may also harm the health of media contacts. Some studies have shown that current media reports tend to attribute obesity to individual behavior without considering other social factors leading to obesity, thus resulting in the prevalence of weight bias, which is prone to adverse effects on the physical and mental health of obese patients [15]. Some studies have also shown that food-sharing and food marketing, which can be seen everywhere in the online media, can induce media contacts to eat more high-calorie and low-nutrient foods [16], which is not conducive to the health of groups with obesity. Although these findings suggest that media use hurts the physical and mental health of people with obesity, some scholars have different results. For example, the use of traditional media, such as television, radio, and newspapers, can positively impact people’s health behavior and prevent negative changes in health-related behavior in large populations [17]. On the one hand, online media can establish social relations, provide social support and improve mental health for the groups in need [18]. On the other hand, it can also affect people’s attitudes and behaviors by transmitting health information, thus having a positive impact on people’s physical and mental health [19]. It shows that media use can improve the physical and mental health of groups with obesity.

In summary, current studies have conducted extensive research and discussion on the impact of different types of media use on the physical and mental health of the population, including groups with obesity, but no unified conclusion can be reached. Accordingly, we raise the following research question:

RQ1. What are the effects of different types of media use on the physical and mental health of people with obesity?

### 1.2. Impacts of Physical Exercise on the Mental Health of People with Obesity

Physical exercise is the performance of some activity to develop or maintain physical fitness and overall health [20]. Many studies have confirmed the positive effect of physical exercise on people’s physical health. It has been deeply described that exercise has a positive effect on physical health, particularly in people with obesity [21,22]. In addition, physical exercise may have a positive impact on the mental health of people with obesity. People with obesity are more prone to depression and anxiety than those with normal weight. There is still evidence that physical exercise may be a neglected mental health intervention. Physical exercise is beneficial to mental health and improves the happiness of exercisers. It can reduce anxiety, depression, and negative emotions and improve self-esteem and cognitive function [23]. A cross-sectional study shows that people who lack physical exercise have a higher risk of depression than those who exercise regularly, and improving mental health through low-intensity or moderate-intensity physical activity is considered an effective means [7]. In summary, existing studies have proved that physical exercise can affect people’s physical and mental health, and this positive impact on groups with obesity may also be noticeable. In this regard, we propose the following research hypothesis:

**Hypothesis** **1.**
*Physical exercise can have a significant positive impact on the mental health of people with obesity.*


### 1.3. Media Use, Physical Exercise, and Physical and Mental Health of People with Obesity

Studies have confirmed a possible relationship between media use and physical exercise. A survey found that social media exercise services can help users control weight and significantly improve their levels of physical activity, health, and perception of happiness [24]. In addition, the research on the influence of media content on people’s physical exercise intention and behavior also provides evidence for establishing the relationship between media use and physical exercise. On the one hand, the media reports on the issue of weight stigma have stimulated the willingness of those who feel obese to exercise. A study of the female population shows that participants with more stigmatization reported stronger fitness intentions: female groups who have access to specific media content through media use generate motivation for physical exercise [25]. On the other hand, the idealized body image created by the media also encourages people to exercise to some extent. Media users enhance their satisfaction with their appearance by strengthening physical exercise to match the idealized body image portrayed by the media [26].

The above findings show a specific correlation between media use, physical exercise, and physical and mental health. A study confirmed that physical exercise played a partial mediating role between media use and perceived health. Media use would significantly positively affect physical activity and positively affect perceived health [27]. However, this study did not distinguish between physical and mental health and did not target the obesity groups. Therefore, whether the results of the mediating effect model are applicable to explain obesity groups has not been involved in the current research. Although the relationship between media use, physical exercise, and physical and mental health of obesity groups has been clear, the potential impact mechanism of the relationship is still unclear. According to the existing literature, this study infers that physical exercise may play an intermediary role between media use and health level and puts forward research hypotheses:

**Hypothesis** **2a.**
*Physical exercise plays an intermediary role in the relationship between Traditional media use and physical and mental health in obesity groups.*


**Hypothesis** **2b.**
*Physical exercise plays an intermediary role in the relationship between online media use and physical and mental health in obesity groups.*


## 2. Materials and Methods

### 2.1. Data Collection

The data used in this study are from the 2017 China General Social Survey (CGSS). CGSS is a national, comprehensive, and continuous academic survey project implemented by the National Survey Research Center at the Renmin University of China. The CGSS survey selects samples by stratified and phased probability sampling method and covers residents of 31 provinces, autonomous regions, and municipalities (except Hong Kong Special Administrative Region, Macao Special Administrative Region, and Taiwan). CGSS sample data in 2017 includes 12,582 valid samples, which is a rare, representative, and comprehensive social survey sample in China. In addition, compared with the previous CGSS data, CGSS 2017 has added the problem of residents’ use of the internet, which facilitates this study to investigate the use of online media in obesity groups. Due to the differences in living standards in different countries and regions of the world, the body mass index (BMI) criteria for obesity groups are also different. In the Chinese population [28], it has long been suggested that the BMI-based definition for obesity showed to be lower than for European or North American populations [29], where obesity is defined as a BMI of 30 kg/m^2^ or greater. The reason for this is [30] because obesity-associated metabolic complications occur at lower BMI in Chinese people compared to in European/North American populations; since obesity is defined as the statistical point at which obesity—associated complication rates accelerate for a population, the definition of obesity for the Chinese population should match the BMI—specific and population—specific complication rates. Overall, the consensus is that BMI cut-offs for obesity showed to be lower in China, whereby obesity was defined as BMI > 27.9 kg/m^2^ [31]. Therefore, in this study, the inclusion criteria for the sample were BMI greater than 27.9 kg/m^2^, and the exclusion criteria for the sample were BMI less than or equal to 27.9 kg/m^2^ or missing core variables (e.g., missing media use, physical exercise, sociodemographic information). After eliminating the data with missing values, 882 valid samples were finally obtained.

### 2.2. Variable Selection

#### 2.2.1. Dependent Variables

In this study, the frequency of self-rated physical health and depression was selected as an indicator to measure the physical and mental health of obese patients.

At the physical level, self-rated health is one of the commonly used indicators to measure the population’s health status [32], which refers to the individual’s subjective perception of their overall health status [33]. Studies have shown that self-rated health measurement is a good indicator of self-rated health [34], consistent with objective health [35]. Therefore, this variable can be used to observe and evaluate the physical health status of obesity groups.

At the psychological level, referring to the previous study [36], the frequency index of depression or frustration is used to measure the mental health level of obesity groups.

The survey items of these two variables in CGSS 2017 data are: Participants were asked to report their physical health status on the five-point scale. Respondents’ answers ranged from “very unhealthy” (=1), “relatively unhealthy” (=2), “average” (=3), “relatively healthy” (=4), and “very healthy” (=5). Participants were asked to report their five-point scale of the frequency of depression or frustration. Respondents’ answers ranged from “never” (=5), “rarely” (=4), “sometimes” (=3), “often” (=2), and “always” (=1).

#### 2.2.2. Independent Variables

This study uses the item ‘Do you often take part in physical exercise in the past year?’ in CGSS to measure the frequency of physical exercise referring to the previous study [37]. (“never” coded as 1, “several times a year or less” coded as 2, “several times a month” coded as 3, “several times a week” coded as 4, and “everyday” coded as 5).

Media use is defined as the extent to which the audience has access to specific information or a particular type of media content [37]. Media use in this study is divided into traditional media use (newspapers, magazines, radio, television) and online media use according to types. In CGSS, participants were asked about the five-point scale that reported their media use in the past year, and the answers ranged from “never” (=1), “rarely” (=2), “sometimes” (=3), “often” (=4), and “always” (=5).

#### 2.2.3. Control Variables

Since previous empirical studies have shown that many demographic variables [38,39,40] are associated with health levels, this study sets some demographic variables to control variables. Individual feature control variables are as follows: (1) Gender (female coded as 1, male coded as 0), (2) Age, (3) Household registration (agriculture account coded as 1, Non-agricultural account coded as 0), (4) Household income in 2016 (The variable is taken as a natural logarithm. Considering that the impact of income on physical and mental health may be non-linear, the square term of annual total household income is added to the regression.), (5) Education level. Participants were asked about their highest education levels in CGSS2017. The variables of educational attainment in this study were recorded as “Uneducated” (=0), “Primary school” (=1), “Middle school” (=2), “High school/Technical secondary school” (=3), “Junior college” (=4), “Bachelor’s degree” (=5), “Master’s degree and above” (=6).

### 2.3. Data Analysis

A total of 882 valid samples were used for data analysis. We checked the outliers and multicollinearity before analysis and recoded and standardized variables to adapt to the research design. This study used descriptive statistics, variance analysis, post hoc analysis, bivariate correlation analysis, and multi-level regression analysis. Descriptive statistical analysis was used to describe the characteristics of education level, media use, and physical and mental health of obesity groups. Variance analysis was used to analyze the differences in physical and mental health and media use frequency among people with obesity with different exercise frequencies. Post hoc analysis was conducted based on ANOVA; used to analyze the relationship between different frequencies of physical exercise and frequency of media use, physical and mental health. Bivariate correlation analysis and regression analysis discussed the effects of exercise frequency and media use on the health of obesity groups. The mediating effect model was used to test the mediating effect of physical exercise on the relationship between media use and physical and mental health. This study used the product coefficient method to test the mediating effect. Since the statistical impact of the product coefficient method is better than that of the causal step method [41], the coefficient product method is used by more and more researchers. The coefficient product method is divided into two categories. One is the Sobel test method based on mediating effect that the sampling distribution is normal. The other is the asymmetric confidence interval method based on the mediating effect that the sampling distribution is non-normal distribution. The asymmetric confidence interval method includes the Bootstrap method and distribution of the Product method. Mackinnon et al. (2004) found that the Bootstrap method had the highest statistical efficacy in mediating effect analysis [42]. Therefore, the Bootstrap sampling method was used to test the dominance in this study. All statistical analyses were performed by SPSS 24.0. The *p*-value is two-sided, and a value less than 0.05 is considered statistically significant.

## 3. Results

### 3.1. Descriptive Statistical Analysis

The demographic information of 882 samples is shown in Table 1. It can be found that the physical health level of obesity groups (M = 3.252, SD = 1.107) is lower than the mental health level (M = 3.787, SD = 1.067). Regarding media use frequency, online media use frequency (M = 2.812, SD = 1.703) in obesity groups is higher than traditional media (M = 2.234, SD = 0.707).

The detailed media use and physical exercise of obesity groups are shown in Table 2. Regarding media use, the frequency of obesity groups using paper, radio, and online media is relatively low. Still, the frequency of exposure to television media is relatively high, and the sum of often and always choices accounts for the vast majority (70.75%). In terms of physical exercise, many obesity groups have never participated in physical exercise (44.78%).

The results of variance analysis show that there are significant differences in exercise frequency between traditional media use frequency, online media use frequency, physical health level, and mental health level (*p* < 0.01). The variance analysis results of exercise frequency and other variables are shown in Table 3. The groups with obesity with the highest frequency of exercise have the highest level of mental health (M = 3.99, SD = 1.10) and the highest frequency of traditional media use (M = 2.57, SD = 0.77). The groups with obesity with often exercise have the highest level of physical health (M = 3.48, SD = 1.12).

It can be found from Table 3 that different physical exercise frequency samples for traditional media use, online media use, physical health, and mental health all showed significance (*p* < 0.05), which means the different physical exercise frequency samples for traditional media use, online media use, physical health, and mental health are different. The post hoc analysis needs to be performed. The results of post hoc analysis are shown in Table 4. The frequency of physical exercise shows a significant level of 0.01 (F = 24.762, *p* = 0.000) for the traditional media use, and the average score comparison results of the groups with significant differences was “Rarely > Never; Sometimes > Never; Often > Never; Always > Never; Always > Rarely”. The frequency of physical exercise shows a significant level of 0.01 (F = 24.113, *p* = 0.000) for online media use, and the average score comparison results of the groups with significant differences were “Rarely > Never; Sometimes > Never; Often > Never; Always > Never”. The frequency of physical exercise shows a significant level of 0.01 (F = 3.433, *p* = 0.009) for physical health, and the average score comparison results of the group with significant differences were “Often > Never”. The frequency of physical exercise shows a significant level of 0.01 (F = 4.132, *p* = 0.003) for mental health, and the average score comparison results of the group with significant differences were “Always > Never”.

### 3.2. Preliminary Analysis

The results of the bivariate correlation analysis are shown in Table 5 and Table 6. Generally speaking, after taking the absolute value of the correlation coefficient (r), 0–0.09 is no correlation, 0.1–0.3 is a weak correlation, 0.3–0.5 is a moderate correlation, and 0.5–1.0 is a strong correlation [43]. The bivariate correlation analysis shows that there is a significant positive correlation between physical health level and the frequency of Online media use and physical exercise frequency (*p* < 0.01). The bivariate correlation analysis shows that there is a significant positive correlation between mental health levels and the frequency of Online media use and physical exercise frequency (*p* < 0.01).

Based on bivariate correlation analysis, further linear regression analysis was performed (Table 7). Model 1 explained 18.1% of the total variance of the outcome variables, with physical health as the dependent variable. Model 2 explained 8.9% of the outcome variables with mental health as a dependent variable. The linear regression results of Model 1 show that the frequency of Online media use (B = 0.067, *p* < 0.05) can significantly positively predict the physical health level of obesity groups. The linear regression results of Model 2 show that the frequency of physical exercise (B = 0.048, *p* < 0.05) can significantly positively predict the mental health level of obesity groups. These results answered RQ1 and showed that hypothesis 1 holds.

### 3.3. Testing for Mediation Effect

To further verify the effect of physical exercise and media use on the physical and mental health of obesity groups, this study selected the frequency of physical exercise as an intermediary variable to explore its possible influencing mechanism on the physical and mental health of obesity groups.

Table 8 is the model involved in mediating effect analysis, including the regression model of independent variables to intermediary variables (Model 4) and the regression model of independent variables to dependent variables (see Model 3 when there is no intermediary variable). Table 9 summarizes the results of the mediating effect model. A represents the regression coefficient of the independent variable to the mediating variable; b represents the regression coefficient of the mediating variable to the dependent variable (Table 7, Model 2); c represents the regression coefficient of independent variables to dependent variables (see Model 3 when there is no intermediary variable), namely the total effect; c’ represents the regression coefficient of the independent variable to the dependent variable (see Model 2 when there is an intermediary variable), namely the direct effect. The bootstrap sampling method tests the intermediary role according to the test procedures. If a and b are significant and c’ is conspicuous, and a * b is synonymous with c’, it is a partial mediating effect. If at least one of a and b is insignificant and a * b’s 95% confidence interval (Boot CI) includes number 0, the mediating effect is insignificant [44].

The test of mediating effect is carried out strictly according to the test steps, and the results that the mediating effect is not obvious are not listed. The results in Table 8 show that physical exercise has a fully mediating effect on the relationship between online media use and mental health. Combined with the previous research, it can be concluded that online media use does not directly affect the mental health of groups with obesity. Still, online media use can improve the mental health of groups with obesity through the intermediary role of physical exercise. These results showed that H2a does not hold and H2b partially holds. The model of the effects of physical exercise and media use on the health of people with obesity is depicted in Figure 1.

## 4. Discussion

This study investigated the effects of different media types and physical exercise on the physical and mental health of groups with obesity by using representative national data. Our study established an intermediary structure model and obtained three valuable findings: (1) Online media use was a possible positive predictor of physical health improvement among people with obesity. (2) Physical exercise was a possible positive predictor of mental health improvement among people with obesity. (3) Physical exercise played a complete mediating role in the relationship between online media use and mental health. This study helps to better understand the relationship between media use, physical exercise, and the health level of groups with obesity.

### 4.1. Online Media Use and Physical Health of Groups with Obesity

The study found that online media use is a possible positive predictor of physical health improvement among people with obesity; traditional media use has nothing to do with the physical and mental health of people with obesity, which differs from previous research. A study shows that traditional and online media use has a significant positive effect on self-rated physical health [27]. We consider that this result may be related to differences in media content of different media types. When the traditional media reports the obesity problem, they do not attribute the obesity problem correctly. Still, they only attribute the responsibility of obesity to individuals rather than other factors, which makes individuals bear the burden of problems that may be caused by society in essence [45]. Mass media coverage of obesity reinforces the stigmatization and stereotyping of people with obesity [46]. It may lead to groups with obesity in contact with traditional media, having difficulty getting effective information about their physical and mental health, and even being negatively affected. Therefore, traditional media use has no significant positive impact on the health of groups with obesity.

Unlike traditional media, with the continuous development of communication technology, online media is deeply integrated into public life, changing people’s lifestyles. In health information, health advocates and the public can better convey the health knowledge and ideas they want to convey by shaping information in diversified media formats rather than being bound by existing media formats [47]. For example, a study on the impact of podcasts on users’ physical health shows that the weight loss content in podcasts has more physiological arousal in participants. The use of podcasts has a greater impact on the audience’s perception of weight loss information. It effectively encourages users to participate in physical exercise, promoting their physical health [48]. In addition, online media lower the threshold of information communication, making information transmission more equitable. In the past, inequality in access to health knowledge was often considered a major obstacle to using information and communication technologies for health [49,50]. Online media provide health information production platforms for health professionals and the public interested in health. Internet users, including obese patients, can more easily access online media and receive guidance from health professionals on health. A survey found that 80 percent of Internet users said they would get health information [51]. It also explains why online media use has a significant positive impact on the physical health of people with obesity. Undeniably, there is also much misinformation related to obesity on the internet. However, since our survey data do not involve media content, we cannot judge the impact of the misinformation on the physical and mental health of obesity. Therefore, we must emphasize that we can only make possible explanations for the results.

### 4.2. Physical Exercise and Mental Health of Groups with Obesity

The second finding of this study was that physical exercise is a possible positive predictor of mental health improvement among groups with obesity. Previous studies have shown that people with obesity are more likely to have psychological problems. Obesity is associated with many mental diseases, including mood disorders, anxiety, personality disorders, overeating, and so on [52]. A study summary shows that obesity has a higher risk of mental illness, ranging from 30% to 70% [53]. There have been many studies on the effects of exercise on mental health. Regular physical exercise can significantly improve mental health and relieve depression, anxiety, and stress symptoms [54]. In a study on obese children, researchers found that implementing physical exercise programs in obese children helped obese children develop positive thinking and improve their emotional health, self-awareness, and self-concept [55]. This study confirms that groups with obesity can positively impact mental health through physical exercise. In addition, it is worth mentioning that we explain the low correlation between the variables in this study and the weak prediction level of the model. We consider that the reason is complex. It has been pointed out that obesity is a complex, multifactorial disease, including genetic and environmental factors [56]. From the perspective of metabolic efficiency of groups with obesity, there may be genetic differences in the metabolism of groups with obesity. Still, there is little evidence that low metabolic rate plays a major role in the development or maintenance of obesity in the vast majority of overweight people, indicating that the development and maintenance of obesity are mediated by the intake of more than normal amounts of food [57]. Excessive food intake requires greater heat consumption, and the generation of heat consumption, in addition to their basic metabolism, also needs a lot of physical exercises to consume more heat. People with obesity are more likely to be obese because of high-calorie food intake, so diet control can have a more positive impact on improving obesity [58]. Therefore, we believe that this may be the reason for the low correlation between physical exercise and mental health variables in this study because the improvement of mental health is not only affected by a single physical exercise factor but may involve more complex factors.

### 4.3. Online Media Use, Physical Exercise, and Mental Health of Groups with Obesity

The most important finding of this study is that for groups with obesity, physical exercise plays a fully mediating role in the use of online media and mental health. In the present society, online media profoundly shapes people’s lifestyles and has a subtle influence on people’s cognition, attitude, and behavior, especially online media can play a more significant role in meeting the psychological needs of users. Benjamin D. Sylveste’s research showed that under high satisfaction of psychological needs, the positive effect of autonomous motivation on perceived diversity of exercise behavior is obvious [59]. The information about physical exercise in the online media affects the cognition of people with obesity and then makes obese patients have the willingness to exercise effectively, and ultimately has a positive impact on their mental health. Previous studies have often failed to consider the underlying mechanisms of health effects of online media use [19]. The empirical results of this study provide new evidence for exploring the potential mechanism of online media use affecting mental health. Therefore, on the one hand, relevant health institutions can strengthen the popularization of physical exercise information on online media so that more people with obesity can effectively access physical exercise information. On the other hand, in terms of innovating health information forms, relevant institutions can make full use of the diversity of online media information forms [60], such as inviting sports stars to drive groups with obesity to physical exercise in the form of live broadcasts and social interaction in the online media, to encourage groups with obesity to carry out physical exercise. In addition, the network platform operators and relevant government departments also need to constantly optimize the network information environment, targeted to solve the possible problems of weight discrimination and obesity stigma in the online media.

### 4.4. Implications and Limitations

First, this study takes the obese group as the starting point. It establishes the basic model of the relationship between media use, physical exercise, and physical and mental health based on this particular group, which has certain theoretical significance for studying the physical and mental health of groups with obesity and exploring the role of media in promoting physical, and mental health. Secondly, the increasing proportion of people with obesity is a serious social public health problem. Through the discussion and research on the relevant issues, this study can provide a way to alleviate the obesity problem, that is, to promote the physical exercise behavior of people with obesity with the help of the media, thereby improving their mental health, which has certain social significance.

In addition, there are still some limitations of this study. First of all, there are subtle differences in the BMI definition criteria for groups with obesity of different ages. This study used the BMI criteria for most Chinese populations, which can be improved in future studies. Secondly, since this study used second-hand data, the measurement items were limited. Hence, we only take media use and physical exercise as critical influencing factors into the study, and future studies can include more other influencing factors. In addition, the second-hand data does not investigate the content of the media use, the methods for physical exercise, the blood parameters of obese patient samples and other patient data, and the data on food consumption. We suggest that future research can avoid this limitation by collecting raw data. Finally, cross-sectional data can only allow us to make possible explanations of the causal relationship between media use, physical exercise, and physical and mental health. Therefore, it is recommended to conduct group studies or further supplement data in the future to assess the causal effects of media use and physical exercise on the physical and mental health of groups with obesity.

## 5. Conclusions

Based on the national representative sample of the CGSS2017, this study explored the effects of different media type use and physical exercise on the health level of groups with obesity. Online media use was a possible positive predictor of physical health improvement among people with obesity. Physical exercise was a possible positive predictor of mental health improvement among people with obesity. Physical exercise played a fully mediating role in the relationship between online media use and mental health. Previous studies have paid less attention to the special group of people with obesity. They also tend to use a single variable to explore the impact of media use or physical exercise on health. By integrating these two variables, this study better explains the influencing mechanism of online media use on the mental health level of groups with obesity. In addition, due to the weak correlation and low model prediction level, we believe that the improvement of physical and mental health in groups with obesity is still a very complex problem. The results of this study can provide references and suggestions for improving the health and well-being of people with obesity.

## Figures and Tables

**Figure 1 healthcare-10-01740-f001:**
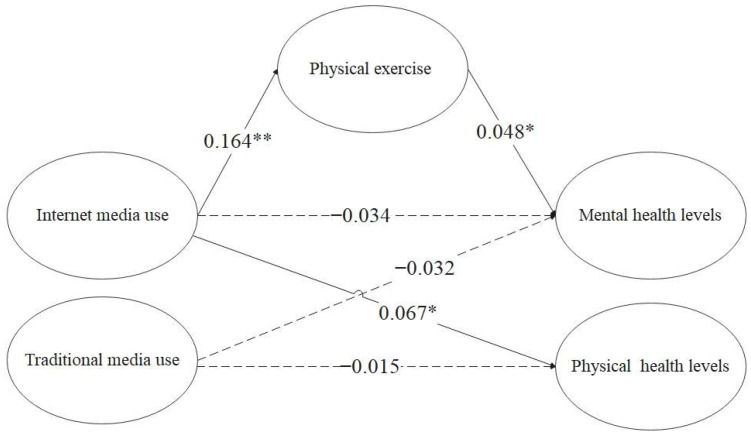
The model of the effects of physical exercise and media use on the health of people with obesity. Notes: * Significant at level *p* < 0.05, ** Significant at level *p* < 0.01.

**Table 1 healthcare-10-01740-t001:** Sociodemographic information of the participants (*n* = 882).

Variable	*n* (%) or Mean ± SD
**Gender**	
Male	422 (47.85)
Female	460 (52.15)
**Age (year)**	51.457 ± 14.929
**Education level**	
Uneducated	89 (10.09)
Primary school	229 (25.96)
Middle school	261 (29.59)
High school/technical secondary school	165 (18.71)
Junior college	75 (8.5)
Bachelor’s degree	58 (6.58)
Master’s degree and above	5 (0.57)
**Household register**	
Agriculture account	448 (50.79)
Non-agricultural account	434 (49.21)
Variable	Median (Interquartile range), Mean ± SD
**physical exercise**	2 (3), 2.542 ± 1.649
**Traditional media use**	2 (1), 2.234 ± 0.707
**Online media use**	3 (4), 2.812 ± 1.703
**Physical health level**	3 (2), 3.252 ± 1.107
**Mental health level**	4 (2), 3.787 ± 1.067

**Table 2 healthcare-10-01740-t002:** Descriptive statistics of participants’ media use and exercise (*n* = 882).

Items	Never *n* (%)	Rarely *n* (%)	Sometimes *n* (%)	Often *n* (%)	Always *n* (%)
1. Newspaper media use	512 (58.05)	200 (22.68)	80 (9.07)	58 (6.58)	32 (3.63)
2. Magazine media use	544 (61.68)	204 (23.13)	95 (10.77)	32 (3.63)	7 (0.79)
3. Broadcast media use	545 (61.79)	144 (16.33)	102 (11.56)	58 (6.58)	33 (3.74)
4. Television media use	36 (4.08)	89 (10.09)	133 (15.08)	333 (37.76)	291 (32.99)
5. Online media use	369 (41.84)	47 (5.33)	69 (7.82)	175 (19.84)	222 (25.17)
6. Exercise	395 (44.78)	117 (13.27)	58 (6.58)	121 (13.72)	191 (21.66)

**Table 3 healthcare-10-01740-t003:** Results of variance analysis.

	Exercise (Mean ± SD)					F	*p*
	Never (*n* = 395)	Rarely (*n* = 117)	Sometimes (*n* = 58)	Often (*n* = 121)	Always (*n* = 191)		
Traditional media use	2.01 ± 0.58	2.30 ± 0.66	2.31 ± 0.73	2.34 ± 0.76	2.57 ± 0.77	24.762	*p* < 0.001 **
Online media use	2.24 ± 1.60	3.18 ± 1.68	3.62 ± 1.36	3.50 ± 1.67	3.08 ± 1.66	24.113	*p* < 0.001 **
Physical health level	3.11 ± 1.16	3.32 ± 1.07	3.33 ± 1.11	3.48 ± 1.12	3.34 ± 0.98	3.433	0.009 **
Mental health level	3.64 ± 1.09	3.79 ± 0.92	3.93 ± 1.02	3.86 ± 1.05	3.99 ± 1.10	4.132	0.003 **

Note: ** *p* < 0.01.

**Table 4 healthcare-10-01740-t004:** Results of post hoc analysis.

	(I) Item	(J) Item	(I) Mean	(J) Mean	Difference value (I–J)	*p*
Traditional media use	Never	Rarely	2.006	2.303	−0.298	0.001 **
	Never	Sometimes	2.006	2.31	−0.305	0.035 *
	Never	Often	2.006	2.339	−0.333	*p* < 0.001 **
	Never	Always	2.006	2.572	−0.566	*p* < 0.001 **
	Rarely	Sometimes	2.303	2.31	−0.007	1
	Rarely	Often	2.303	2.339	−0.035	0.997
	Rarely	Always	2.303	2.572	−0.269	0.021 *
	Sometimes	Often	2.31	2.339	−0.028	0.999
	Sometimes	Always	2.31	2.572	−0.262	0.15
	Often	Always	2.339	2.572	−0.233	0.064
Online media use	Never	Rarely	2.243	3.179	−0.936	*p* < 0.001 **
	Never	Sometimes	2.243	3.621	−1.378	*p* < 0.001 **
	Never	Often	2.243	3.504	−1.261	*p* < 0.001 **
	Never	Always	2.243	3.079	−0.835	*p* < 0.001 **
	Rarely	Sometimes	3.179	3.621	−0.441	0.579
	Rarely	Often	3.179	3.504	−0.325	0.665
	Rarely	Always	3.179	3.079	0.101	0.991
	Sometimes	Often	3.621	3.504	0.117	0.995
	Sometimes	Always	3.621	3.079	0.542	0.29
	Often	Always	3.504	3.079	0.426	0.277
Physical health	Never	Rarely	3.109	3.316	−0.207	0.525
	Never	Sometimes	3.109	3.328	−0.219	0.737
	Never	Often	3.109	3.479	−0.37	0.034 *
	Never	Always	3.109	3.34	−0.231	0.225
	Rarely	Sometimes	3.316	3.328	−0.011	1
	Rarely	Often	3.316	3.479	−0.163	0.86
	Rarely	Always	3.316	3.34	−0.024	1
	Sometimes	Often	3.328	3.479	−0.152	0.946
	Sometimes	Always	3.328	3.34	−0.013	1
	Often	Always	3.479	3.34	0.139	0.881
Mental health	Never	Rarely	3.641	3.795	−0.154	0.751
	Never	Sometimes	3.641	3.931	−0.291	0.434
	Never	Often	3.641	3.86	−0.219	0.413
	Never	Always	3.641	3.995	−0.354	0.006 **
	Rarely	Sometimes	3.795	3.931	−0.136	0.958
	Rarely	Often	3.795	3.86	−0.065	0.994
	Rarely	Always	3.795	3.995	−0.2	0.63
	Sometimes	Often	3.931	3.86	0.072	0.996
	Sometimes	Always	3.931	3.995	−0.064	0.997
	Often	Always	3.86	3.995	−0.135	0.877

Note: * *p* < 0.05; ** *p* < 0.01.

**Table 5 healthcare-10-01740-t005:** Bivariate correlation between physical health level and other variables.

	Physical Health Level	Gender (Male)	Age	Household Register (Agriculture)	Total household Income	Education Level	Traditional Media Use	Online Media Use
Gender(female)	−0.155 **							
Age	−0.318 **	0.142 **						
Household register (Agriculture)	−0.038	0.074 *	−0.150 **					
Total household income	0.319 **	−0.147 **	−0.195 **	−0.336 **				
Education level	0.253 **	−0.292 **	−0.381 **	−0.454 **	0.493 **			
Traditional media use	0.015	−0.144 **	0.198 **	−0.288 **	0.192 **	0.258 **		
Online media use	0.323 **	−0.198 **	−0.561 **	−0.263 **	0.425 **	0.597 **	0.089 **	
Exercise	0.104 **	−0.109 **	0.012	−0.327 **	0.192 **	0.322 **	0.306 **	0.239 **

Note: * *p* < 0.05; ** *p* < 0.01.

**Table 6 healthcare-10-01740-t006:** Bivariate correlation between mental health level and other variables.

	Mental Health Level	Gender (Male)	Age	Household Register (Agriculture)	Total Household Income	Education Level	Traditional Media Use	Online Media Use
Gender(female)	−0.113 **							
Age	−0.065	0.142 **						
Household register (Agriculture)	−0.103 **	0.074 *	−0.150 **					
Total household income	0.250 **	−0.147 **	−0.195 **	−0.336 **				
Education level	0.235 **	−0.292 **	−0.381 **	−0.454 **	0.493 **			
Traditional media use	0.064	−0.144 **	0.198 **	−0.288 **	0.192 **	0.258 **		
Online media use	0.135 **	−0.198 **	−0.561 **	−0.263 **	0.425 **	0.597 **	0.089 **	
Exercise	0.131 **	−0.109 **	0.012	−0.327 **	0.192 **	0.322 **	0.306 **	0.239 **

Note: * *p* < 0.05; ** *p* < 0.01.

**Table 7 healthcare-10-01740-t007:** Linear regression analysis results.

	Model 1				Model 2			
	B	SE	β	95% CI	B	SE	β	95% CI
Constant	2.459 **	0.311		1.849~3.068	2.498 **	0.314		1.883~3.112
Age	−0.014 **	0.003	−0.185	−0.020~−0.007	0.002	0.003	0.028	−0.004~0.008
Gender (female)	−0.165 *	0.072	−0.075	−0.306~−0.024	−0.097	0.073	−0.046	−0.239~0.046
Household register (Agriculture)	0.15	0.087	0.068	−0.021~0.322	0.096	0.088	0.045	−0.077~0.269
Total household income	0.06 **	0.009	0.249	0.042~0.077	0.044 **	0.009	0.192	0.027~0.061
Education level	−0.009	0.038	−0.011	−0.084~0.065	0.134 **	0.038	0.173	0.059~0.209
Traditional media use	−0.015	0.055	−0.01	−0.123~0.093	−0.032	0.056	−0.021	−0.141~0.077
Online media use	0.067*	0.029	0.103	0.009~0.124	−0.034	0.03	−0.054	−0.092~0.024
Exercise	0.036	0.023	0.053	−0.010~0.081	0.048 *	0.023	0.074	0.002~0.094
R^2^	0.181				0.089			
Adjusted R^2^	0.173				0.08			
F	F (8852) = 23.521, *p* = 0.000				F (8852) = 10.376, *p* = 0.000			
Dependent variable	Physical health level				Mental health level			

Note: * *p* < 0.05; ** *p* < 0.01, B (Unstandardized Beta), SE (standard error) β (Standardized Beta), CI (confidence interval).

**Table 8 healthcare-10-01740-t008:** Results of Mediating Effect Analysis.

	Model 3				Model 4			
	B	SE	β	95% CI	B	SE	β	95% CI
Constant	2.519 **	0.31	-	1.912~3.127	1.037 *	0.46	-	0.136~1.938
Age	0.002	0.003	0.033	−0.004~0.008	0.018 **	0.005	0.162	0.009~0.027
Gender (female)	−0.097	0.072	−0.046	−0.239~0.045	−0.078	0.107	−0.024	−0.289~0.133
Household register (Agriculture)	0.072	0.087	0.034	−0.099~0.243	−0.557 **	0.13	−0.169	−0.811~−0.303
Total household income	0.044 **	0.009	0.19	0.026~0.061	−0.004	0.013	−0.01	−0.029~0.022
Education level	0.141 **	0.037	0.182	0.068~0.215	0.247 **	0.055	0.205	0.139~0.356
Online media use	−0.027	0.029	−0.043	−0.084~0.031	0.164 **	0.044	0.169	0.078~0.249
R^2^	0.084				0.166			
Adjusted R^2^	0.078				0.161			
F	F (6854) = 13.106, *p* = 0.000				F (6854) = 28.429, *p* = 0.000			
Dependent variable	Mental health level				Exercise			

Note: * *p* < 0.05; ** *p* < 0.01, B (Unstandardized Beta), SE (standard error) β (Standardized Beta), CI (confidence interval).

**Table 9 healthcare-10-01740-t009:** Results of Mediating effects.

Item		Online Media Use ≥ Exercise ≥ Mental Health Level
c	Total effect	−0.027
a		0.164 **
b		0.045
a * b	Mediating effect value	0.007
a * b	(Boot SE)	0.007
a * b	(z value)	0.994
a * b	(*p*-value)	0.32
a * b	(95% BootCI)	0.000~0.030
c’		−0.034
Test result	Direct mediating effect	Complete mediating effect
Effect account		100%

Note: * *p* < 0.05; ** *p* < 0.01.

## Data Availability

Publicly available datasets were analyzed in this study. This data can be found here: http://cgss.ruc.edu.cn/English/Home.htm (accessed on 1 May 2022).

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
