# Peer review of "Impacts of Physical Exercise and Media Use on the Physical and Mental Health of People with Obesity: Based on the CGSS 2017 Survey"

_healthcare, 2022, doi:10.3390/healthcare10091740_

Round 1

Reviewer 1 Report

The manuscript by Han Wang., et al., entitled as “Effects of physical exercise and media use on the physical and mental health of people with obesity: Based on the CGSS 2017 Survey” analyzed the media use and physical exercise effects on obese people mental health. Authors have used data from 2017 China general social survey to model the findings. Here are my comments

1.     Table 2 shows, 44% of participants never do exercise, 41% never use internet media, Did authors find these subjects obesity level. Are they have severe obesity?

2.     How have authors validated mental health based on this survey?

3.     What are the methods for physical exercise? At home or gym? Aerobic or resistance training? What have they preferred?

4.     Obese patient samples (blood parameters) would add more value to this manuscript or other patient data could help us to understand better. Do the authors have access to patient samples?

5.     As the obese people have overeating issues, do authors have any data from survey about food consumption? What type of food they prefer and eat more? Do they change their eating habits after using media or physical exercise? Does their education and economic standards limiting their change in food habits?

Author Response

Dear reviewer,

Many thanks to you for providing comments on this manuscript. We believe that the comments are constructive in enhancing the manuscript. And we have carefully considered the comments and have made detailed changes to the manuscript with the tracked changes. The following are our responses to your comments.

Reply to the comments

1. Table 2 shows, 44% of participants never do exercise, 41% never use internet media, Did authors find these subjects obesity level. Are they have severe obesity?

Response: Thank you for your comments. As we mentioned in the 2.1. Data Collection (lines182-185), the inclusion criteria of the subjects analyzed in this article are BMI > 27.9. Therefore, all participants in Table 2 suffered from obesity.

2. How have authors validated mental health based on this survey?

Response: Thank you for your comments. For validation of mental health, we presented in 2.2.1. Dependent variables (lines197-198): “At the psychological level, according to the relevant literature, the frequency index of depression or frustration is used to measure the mental health level of obesity groups.”

3. What are the methods for physical exercise? At home or gym? Aerobic or resistance training? What have they preferred?

Response: Thank you for your comments. We used second-hand data for analysis and the original survey did not ask for specific details about participation in physical exercise. CGSS only measured the exercise frequency of participants through the item ' Do you often take part in physical exercise in the past year? ' In the new manuscript, we have added this limitation to section 4.4. Implications and limitations.

4. Obese patient samples (blood parameters) would add more value to this manuscript or other patient data could help us to understand better. Do the authors have access to patient samples?

Response:Thank you for your comments. The data we used for analysis was second-hand. The blood parameters of obese patients and other patient data were not investigated. In the new manuscript, we have added this limitation to section 4.4. Implications and limitations.

5. As the obese people have overeating issues, do authors have any data from survey about food consumption? What type of food they prefer and eat more? Do they change their eating habits after using media or physical exercise? Does their education and economic standards limiting their change in food habits?

Response:Thank you for your comments. The data we used for analysis was second-hand. The data on food consumption were not investigated. In the new manuscript, we have added this limitation in section 4.4. Implications and limitations. In addition, our main research objective is to investigate the impact of media use and physical exercise on physical and mental health. Eating habits are not the main research objective of this paper. Your question gives us inspiration. In future research, we will try to investigate the association between media use, physical exercise, socioeconomic status and eating habits of people with obesity.

Reviewer 2 Report

Dear authors,

Wang et al. investigated the influence of media use on physical and mental health in a sample of people with obesity from China and tried to clarify the influence of physical exercise in these relationships.

I think that the study tries to strike a relevant topic, however, I have several concerns about how the authors interpret their results and the potential conclusions that can be derived from this study. Here are my comments in detail:

Abstract

Please replace "obese people" for "people with obesity" throughout the manuscript. That vocabulary is not appropriate.

In the conclusion section of the abstract, please clarify to which two influencing factors are referring to.

Introduction

Overall, in my opinion is a lenghty section that can be shortened, particularly in the section of the effects of exercise on physical health, which is a topic that has been throughfully described previously.

Lane 60-61: please replace: "on the one hand", for "on one hand".

Lane 102-103: please replace "...we raise research questions" for "...we raise the following research question".

Hypothesis 1a: it has been deeply described that exercise has a positive effect on physical health, particularly in people with obesity.

It is unclear which is the main research question of this study.

Methods

Please expand on why you set the standard for obesity at 27.9 hg/m2.

Please don't use the concept "relevant literature" as its quite subjective. Please explain why these measurement indexes were use for your outcomes.

Considering that the content of the media use (e.g. health from reliable sources or just personal opinions, politics, etc) is closely linked to the effect that it has on mental health, the absence of this information is a big limitation for this study. 

Results

Considering the non-continupus nature of some of your outcomes (e.g. personal and mental health), they should be expressed in terms of median and interquartile range. 

In table 3, please add a post hoc analysis to see between which groups the differences are.

In table 4 and 5, please add the outcome names in the same row for clarity. And also, please differentiate the statistical strength of the correlation coefficients that you calculated, assuming, for example, that a correlation coefficient lower than 0.3 indicates a weak association.

In table 6, the R2 are very low, which indicates that the model that you calculated does not give a good prediction. Please add this to the discussion.

Discussion

In terms of the effect of the internet use on physical health, a significant amount of information related to obesity and health is false or incorrect. Please add this factor in the discussion as well.

Lane 363-364: how your results support the statement that physical exercise has no positive impact on physical health if table 3 shows the opposite?

Conclusion

The conclusion is misleading. Particularly because the correlation strength and prediction level of the models proposed here are low and weak respectively. This should be taken into account in this section.

Author Response

Dear reviewer,

Many thanks to you for providing comments on this manuscript. We believe that the comments are constructive in enhancing the manuscript. And we have carefully considered the comments and have made detailed changes to the manuscript with the tracked changes. The following are our responses to your comments.

Reply to the comments

1. Abstract-Please replace "obese people" for "people with obesity" throughout the manuscript. That vocabulary is not appropriate.

Response: Thank you for your advice. I'm sorry for the inaccurate vocabulary in our manuscripts. In the new manuscript, we have replaced 'obese people' with 'people with obesity'.

2. Abstract-In the conclusion section of the abstract, please clarify to which two influencing factors are referring to.

Response: Thank you for your comments. I am sorry for our negligence. The two factors are media use and physical exercise. To avoid misunderstandings, we have changed the statement in the new manuscript: “Future research could explore more aspects of physical activity (e.g. the methods for physical exercise) and media use (e.g. media preference or compulsive use).”

3. Introduction-Overall, in my opinion is a lenghty section that can be shortened, particularly in the section of the effects of exercise on physical health, which is a topic that has been throughfully described previously.

Response: Thank you for your advice. We fully agree with your advice. In the new manuscript, we have deleted the part about the impact of exercise on physical health.

4. Lane 60-61: please replace: "on the one hand", for "on one hand".

Response: Thank you for your advice. In the new manuscript, we have replaced 'on the one hand ' with 'on one hand '.

5. Lane 102-103: please replace "...we raise research questions" for "...we raise the following research question".

Response: Thank you for your advice. In the new manuscript, we have replaced '... we raise research questions ' with '... we raise the following research question '.

6. Hypothesis 1a: it has been deeply described that exercise has a positive effect on physical health, particularly in people with obesity. It is unclear which is the main research question of this study.

Response: Thank you for your advice. We have deleted hypothesis 1a and the related literature in the new manuscript. Some clarification is in order, the main goal of this study is to verify Hypothesis 2, but Research Question 1 and Research Hypothesis 1 are also very important to our research. We believe that only by studying the effects of media use or physical activity on health can we further address our main research questions: “Whether physical exercise may play a mediating role between media use and health level.”

7. Methods-Please expand on why you set the standard for obesity at 27.9 hg/m2.

Response: Thank you for your advice. Our new manuscript (lines 173-185) explains why this study sets the obesity standard at 27.9g / m2.

8. Methods-Please don't use the concept "relevant literature" as its quite subjective. Please explain why these measurement indexes were use for your outcomes.

Response: Thank you for your advice. In the new manuscript, we have deleted the concept of 'relevant literature'. The measurement indicators in this study are selected with reference to previous studies.

9. Methods-Considering that the content of the media use (e.g. health from reliable sources or just personal opinions, politics, etc) is closely linked to the effect that it has on mental health, the absence of this information is a big limitation for this study. 

Response: Thank you for your advice. We strongly agree that the lack of information on media usage is a big limitation of this study. The data we use for analysis is secondary, so there is no survey data on media usage. In the new manuscript, we have added this limitation to section 4.4. Implications and limitations.

10. Results-Considering the non-continupus nature of some of your outcomes (e.g. personal and mental health), they should be expressed in terms of median and interquartile range. 

Response: Thank you for your advice. We have modified Table 1 and added the median and interquartile range.

11. Results-In table 3, please add a post hoc analysis to see between which groups the differences are.

Response: Thank you for your advice. Following Table 3, we newly add Table 4-Results of post hoc analysis and explain the results.

12. Results-In table 4 and 5, please add the outcome names in the same row for clarity. And also, please differentiate the statistical strength of the correlation coefficients that you calculated, assuming, for example, that a correlation coefficient lower than 0.3 indicates a weak association.

Response: Thank you for your advice. We have revised the form. In the new manuscript, we have added the outcome names on the same row and differentiated the statistical strength of the correlation coefficients.

13. Results-In table 6, the R2 are very low, which indicates that the model that you calculated does not give a good prediction. Please add this to the discussion.

Response: Thank you for your advice. We give a possible explanation for the weak prediction level of the model.

14. Discussion-In terms of the effect of the internet use on physical health, a significant amount of information related to obesity and health is false or incorrect. Please add this factor in the discussion as well.

Response: Thank you for your advice. We have added this factor to the discussion (lines 393-397): “Undeniably, there is also much misinformation related to obesity on the internet. However, since our survey data do not involve media content, we cannot judge the impact of the misinformation on the physical and mental health of obesity. Therefore, we must emphasize that we can only make possible explanations for the results.”

15. Discussion-Lane 363-364: how your results support the statement that physical exercise has no positive impact on physical health if table 3 shows the opposite?

Response: Thank you for your comments. We strongly agree with what you say “it has been deeply described that exercise has a positive effect on physical health, particularly in people with obesity.” Therefore, we have deleted hypothesis 1a, the literature review and this misleading conclusion in the new manuscript. In addition, we have explained the possible reasons for the low correlation and the weak prediction level of the model.

16. Conclusion-The conclusion is misleading. Particularly because the correlation strength and prediction level of the models proposed here are low and weak respectively. This should be taken into account in this section.

Response: Thank you for your advice. We have rewritten the conclusion to avoid the causal explanation.

Reviewer 3 Report

Dear authors,

I have read the manuscript tiled (Effects of physical exercise and media use on the physical and 2 mental health of people with obesity: Based on the CGSS 2017 3 Survey), and I found it worth for publication because it dealt with problems (obesity, physical exercise, media usage, mental and physical health) which are considered as major health concern in the world. However, I have some issues related to this manuscript.

Introduction

You have made extensive review in the literature related to obesity, definition, consequences, effect of media and physical exercise on health among obese people, as well as media, physical exercise, and physical and mental health of obese people.

I think that extensive review and too many references are not required, and some cited references are not relevant to your research. I suggest reducing your introduction and references and to focus mainly on providing fair background and explaining the gap in literature.

Methods:

Did the survey provided inclusion and exclusion criteria? if so, please mention them.

Results:

The results are well explained in the tables; however, I suggest adding some figures.

P <0.000, better to express it like P <.001.

Table 4 and 5 are not clear to me and does not fit well.

Table 6. As linear regression applied, I suggest adding both (Unstandardized Beta, as well as Standardized Beta). Also, significant level and confidence interval are required to be added in the table of linear regression.

Footnotes need to be added for all tables where required. Mainly for tables (6-7-8).

You mentioned in the results section (This study used the product coefficient method to test the mediating effect) (The coefficient product method is divided into two categories. One is the Sobel test method) (The other is the asymmetric confidence interval method) (The asymmetric confidence interval method in- 291 cludes the Bootstrap method and distribution of the Product method). All these details need to be shifted to the data analysis section. Results section should only be focused on your results.

Discussion:

The discussion is well written and supported the results.

Author Response

Dear reviewer,

Many thanks to you for providing comments on this manuscript. We believe that the comments are constructive in enhancing the manuscript. And we have carefully considered the comments and have made detailed changes to the manuscript with the tracked changes. The following are our responses to your comments.

Reply to the comments

1. Introduction-You have made extensive review in the literature related to obesity, definition, consequences, effect of media and physical exercise on health among obese people, as well as media, physical exercise, and physical and mental health of obese people. I think that extensive review and too many references are not required, and some cited references are not relevant to your research. I suggest reducing your introduction and references and to focus mainly on providing fair background and explaining the gap in literature.

Response: Thank you for your advice. We have streamlined the Introduction section in our new manuscript. In addition, we have deleted research hypothesis 1a and related literature.

2. Methods-Did the survey provided inclusion and exclusion criteria? if so, please mention them.

Response: Thank you for your advice. We have supplemented the inclusion and exclusion criteria for the sample (lines173-185).

3. Results-The results are well explained in the tables; however, I suggest adding some figures.

Response: Thank you for your advice. We highly recognize that manuscripts need to add pictures. Therefore, we have added Figure 1 to the new manuscript.

4. P <0.000, better to express it like P <.001.

Response: Thank you for your advice. Your suggestion has taught us a lot. This problem has been revised in the new manuscript.

5. Table 4 and 5 are not clear to me and does not fit well.

Response: Thank you for your advice. In the new manuscript, we have revised two tables.

6. Table 6. As linear regression applied, I suggest adding both (Unstandardized Beta, as well as Standardized Beta). Also, significant level and confidence interval are required to be added in the table of linear regression.

Response: Thank you for your advice. In the new manuscript, we have added non-standardized Beta, standardized Beta, significance levels, and confidence intervals.

7. Footnotes need to be added for all tables where required. Mainly for tables (6-7-8).

Response: Thank you for your advice. In the new manuscript, we have added note to the table.

8. You mentioned in the results section (This study used the product coefficient method to test the mediating effect) (The coefficient product method is divided into two categories. One is the Sobel test method) (The other is the asymmetric confidence interval method) (The asymmetric confidence interval method in- 291 cludes the Bootstrap method and distribution of the Product method). All these details need to be shifted to the data analysis section. Results section should only be focused on your results.

Response: Thank you for your advice. In the new manuscript, we have adjusted these details to the data analysis section.

Round 2

Reviewer 2 Report

Dear authors,

thank you for kindly considering my comments and suggestions in your manuscript.

As a minor comment, please change the title of Table 6 from "physical" to "mental health".

Author Response

Dear Reviewer, Thank you for pointing out a minor error in our manuscript. We have revised this error in the new manuscript. Please forgive our oversight; we have double-checked the manuscript to ensure that it meets the requirements for acceptance.